# Some hidden traps of confidence intervals in medical image segmentation: coverage issues

**Abstract.** Medical imaging AI models are usually assessed by reporting an empirical summary statistic of the performance metric, most commonly the mean or median. Recent work has shown that most studies overlook the uncertainty of these estimates, potentially leading to misleading conclusions and hampering clinical translation of medical imaging AI models. To address this issue, systematic reporting of confidence intervals (CIs) has been recommended, but numerous different CI methods exist, and there is very little literature on their behavior in medical imaging. A fundamental property of a CI method is its coverage. This paper contributes towards filling this literature gap in the context of medical image segmentation, studying the coverage of five CI methods for the two arguably most common summary statistics, the mean and the median. To that purpose, we perform a large-scale analysis of CI coverage using non-parametric simulations based on benchmarks instances representing diverse real-world distributions of two common segmentation metrics (Dice similarity coefficient and normalized surface distance). For the mean, all CI methods have decent coverage for most instances when sample sizes exceed 50, even though there are exceptions. For CIs of the median, we unveil major pitfalls: two common bootstrap CI methods have a catastrophic behavior on average whereas another only fails on very degenerate distributions. We believe these pitfalls are important to communicate to the community and that these findings will contribute to future efforts to provide standardized guidelines on confidence interval reporting in medical imaging AI.

**Keywords:** Medical imaging · Validation · Confidence intervals · Segmentation.

## 1 Introduction

The new FDA guidelines for AI devices [11] require to report not only a summary statistic of the performance but also a measure of uncertainty. Current practice in medical imaging AI is to only report an empirical summary statistic of the performance (most often the mean, sometimes the median), without assessing how precise this estimate is. For example, a recent study found that the majority of segmentation papers from MICCAI 2023 did not assess performance variability at all, while only a single paper reported confidence intervals (CI) [6]. They

further showed that such practices can sometimes be highly misleading and lead to spurious conclusions. These practices may also contribute to little industrial and clinical translation of academic research. Overall, reporting CIs is thus highly recommended.

However, different methods exist for computing CIs. The most popular methods can broadly be divided into parametric and non-parametric methods. Parametric methods rely on assumptions on the distribution of the data. Non-parametric methods are mainly different types of bootstrap.

We are not aware of any study that compares the adequateness of different CIs methods in medical imaging AI, a situation which is different from other research fields such as psychology [18] or economics [5] for instance. In generalist AI, a major company just released guidelines on reporting variability [15]. The formula they provide assumes that the test set is large and is only applicable when the summary statistic is the mean. In medical imaging AI, sample sizes often range from small to moderate. Furthermore, there are cases where the mean is not the most suited summary statistic, for instance in the presence of outliers or skewness, and other statistics, such as the median, need to be reported.

Different properties guide the choice of a given CI method, including coverage and width. The first property to assess is coverage. In particular, CI methods with low coverage lead to overconfidence in the precision of performance estimates. Then, among CI methods with adequate coverage, one can look at other properties such as width.

This work aims at studying the coverage of different CI methods for the arguably two most common summary statistics: the mean and the median. Specifically, we perform a large-scale analysis of 228 benchmarking instances, covering 12 segmentation tasks and 19 algorithms, for two performance metrics, the Dice Similarity Coefficient (DSC) and the Normalized Surface Difference (NSD). For each instance, we analyze the behavior of CIs using non-parametric simulations that adequately represent real-life scenarii. We unveil pitfalls which are of practical importance for the community.

## 2   Methods

### 2.1   Confidence interval methods

A CI is a way to provide information about the precision of a summary statistic. A X% CI method for a statistic is a procedure which generates intervals such that, when multiple sets are drawn from the same distribution and one computes an interval for each set, X% of the intervals will contain the true value of the statistic. For instance, considering a 95% CI method to estimate the mean (resp. median) of a distribution, if we take 100 sets from this distribution, 95 intervals should contain the true mean (resp. median). The proportion of CIs that should contain the true value of the statistic is called theoretical coverage, whereas the observed proportion is called empirical coverage.

CI methods can be broadly divided into two categories: parametric and non-parametric methods. Parametric methods provide theoretical fixed-sample guarantees about the empirical coverage using assumptions on the distribution of the data. The most common non-parametric methods are arguably variations of the bootstrap. They give less guarantees about empirical coverage than the parametric methods, but require no assumption on the distribution of the data [8].

In this paper, we studied five CI methods, which we perceive to be particularly frequently used: two parametric methods using normality assumptions, and the three bootstrap methods implemented in SciPy [19], considering that our community is mostly Python-based [9]. A detailed description of each method can be found in [8], but we will briefly recall them here. The two parametric methods are "parametric t" and "parametric z". They are both based on normality assumptions. "Parametric z" supposes the true variance is known (in practice this is reasonable when the sample size is large), whereas "parametric t" makes a correction for variance estimation ("parametric t" tends to "parametric z" when sample size goes to infinity). The studied bootstrap methods are "percentile", "basic" (also known as reverse percentile) and "BCa" (bias-corrected and accelerated). "Percentile" works by computing bootstrap sets and the statistic of interest for each set, then taking the quantiles of the bootstrap distribution to form a CI. "Basic" aims at reducing the bias of the bootstrap distribution. One computes the bootstrap distribution of the difference to the estimated statistic and a CI is given by mean $\pm$ quantiles of the difference. "BCa" aims to correct for bias and skewness in bootstrap distributions. It is based on a bias coefficient and an acceleration coefficient which uses a jackknife estimate of the statistic of interest to correct for skewness. "BCa" is the default method in SciPy.

Note that, beyond making no assumptions about the metric's distribution, bootstrap methods offer the advantage of being applicable to a wide range of summary statistics, unlike "parametric t" or "parametric z".

### 2.2   Dataset

In this study, we used the Medical Segmentation Decathlon (MSD) [3] challenge which features a wide range of 17 diverse tasks across 10 different organs along with model performance for a large set of 19 models. Details about tasks and models can be found in the MSD paper [3]. To ensure statistical robustness, we selected only the 12 subtasks with test sets containing more than 50 3D images. Test set sizes vary between 59 and 263 (median=139). During the challenge, 19 different models were submitted, and tested on each subtask independently, amounting to 12 tasks $\times$ 19 models = 228 different benchmarking instances. For each instance, we analyzed the values of the Dice Similarity Coefficient (DSC) and the Normalized Surface Distance (NSD) across all test set 3D images.

### 2.3   Assessing confidence interval methods

The purpose of our analysis was to investigate the following research question: do the different CI methods have adequate coverage? For a CI method to be

adequate, its empirical coverage needs to be close to its theoretical coverage. If the coverage is too low, the user is misled to believe that the estimates are more precise than they actually are.

However, empirical coverage computation requires simulating from a distribution because the ground truth value of the statistic must be known. In our case, the distribution would be that of the segmentation metric for a given benchmarking instance. We thus first need to fit a distribution to each of the benchmarking instances. We then perform simulations by sampling from the fitted distribution, for each instance. It is essential that these simulations reflect the reality of the data. Across the literature, these simulations are most often performed under parametric assumption (e.g. [13,14,16]). However, it is unknown if segmentation metrics follow a parametric distribution. We assessed this by performing Kolmogorov-Smirnov tests to see if our distributions matched any of a wide variety of common parametric distributions, including Normal, SkewNormal, 1-LogNormal, 1-Exponential, Beta and Logistic distributions. As shown in the results, none of these parametric distributions was an adequate fit for DSC nor NSD. Therefore, parametric simulations would not correspond to the reality of these segmentation metrics and non-parametric simulations need to be used instead. To achieve this, for each of the 228 instances and for the two metrics (DSC and NSD), we first estimated the underlying metric distribution using kernel density estimation (KDE) with the Epanechnikov (parabolic) kernel. This peculiar kernel choice was made following [20]. It provides a concentrated and smooth interpolation around data points, thus keeping the interpolated distribution close to the original data. To tackle the problem of our bounded metric, we used an adaptive bandwidth to keep all the mass inside the domain. From each of the $2 \times 228$ KDE distributions, we drew 10000 sets (i.e. random realizations of the distribution) of varying size $n$, thus resulting in $2 \times 228$ experiments. We call "instance" the actual metric distribution, and "experiment" the analysis and simulations performed on each instance. For each set, we computed the corresponding CI for each method. Across the 10000 sets, we computed the proportion of CIs containing the true value of the statistic (mean or median), i.e the empirical coverage.

We repeated this process for $n = 10, 25, 50, 75, 100, 125, 150, 200, 250$, to place ourselves in regimes close to those present in the MICCAI papers. Indeed, we meta-analyzed the segmentation papers published at MICCAI 2023 and found that the median test set size was 62 (IQR: 25–223).

## 3   Results

We first briefly describe the results on parametric fits before moving to our main results on CI, which correspond to the core research question of this paper. DSC and NSD are in general non-normal: the Gaussian distribution was rejected in 78% (181/228) of cases for DSC and in 86% for NSD (198/228). Other distributions were rejected between 27% and 86% of cases. A realistic simulation had therefore to be non-parametric.

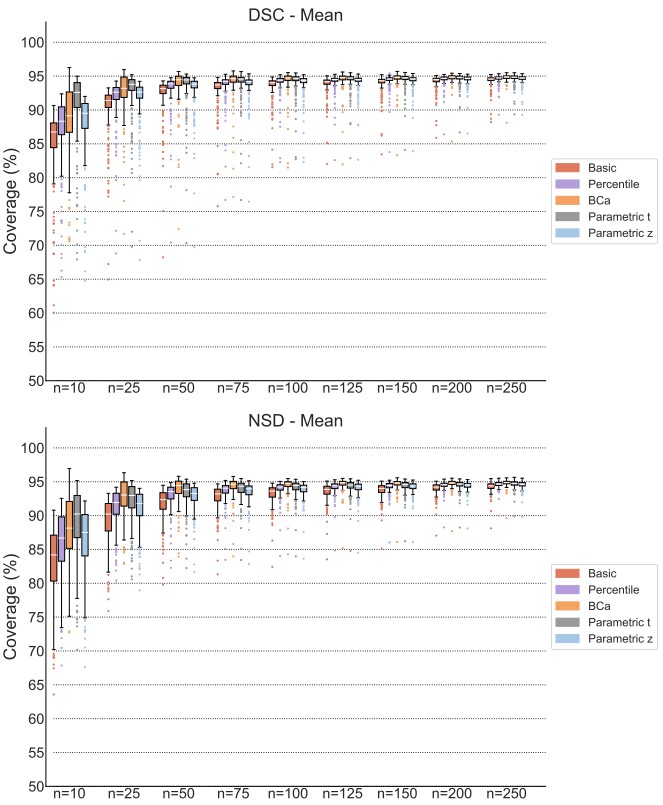

**Fig. 1. Coverage of CIs of the mean. All studied methods have adequate coverage for the majority of instances when $n$ is larger than 50. But there are exceptions with some instances exhibiting low coverage.** For smaller values of $n$, "parametric t" and "BCa" have better coverage. For both the DSC (top) and the NSD (bottom), coverage is shown for each of the tested methods, as a function of the test set size. The boxplots represent the distribution of coverage across experiments. Boxes correspond to median and inter-quartile range (IQR) while whiskers correspond to 2.5 and 97.5 percentiles. Points represent outliers.

The behavior of CIs is presented in Figures 1 (CIs of the mean) and 2 (CIs of the median). Behaviors were similar for both DSC and NSD.

For CIs of the mean, all methods behave decently across the majority of experiments when the test set size is larger than 50. However, there are exceptions: a few instances have low coverage for all methods. For sets smaller than 25, "parametric t" and "BCa" exhibit better coverage. "Basic" performs systematically worse than others, even though the difference is minimal when $n$ is large.

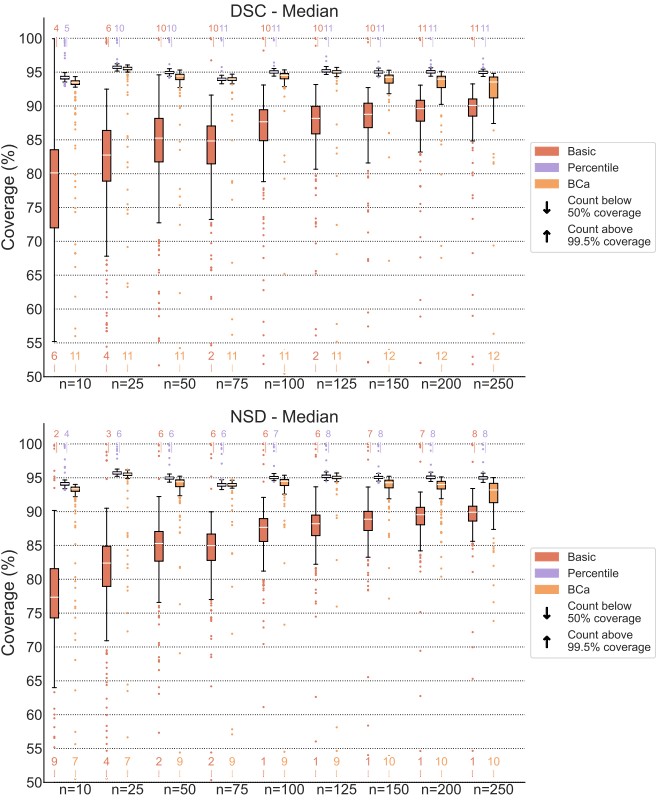

**Fig. 2. Coverage of CIs of the median. Percentile bootstrap is adequate across all $n$ and all experiments. Basic fails dramatically. BCa degrades as n increases, and produces catastrophic failures.** For both the DSC (top) and the NSD (bottom) coverage is shown for each of the tested methods, as a function of the test set size. The boxplots represent the distribution of coverage across experiments. Boxes correspond to median and inter-quartile range (IQR) while whiskers correspond to 2.5 and 97.5 percentiles. Points represent outliers.

For CIs of the median, we unveiled that "basic" bootstrap performs catastrophically for a large number of experiments. "BCa" bootstrap degrades as n increases, and exhibits huge undercoverage for a rather large number of instances. On the other hand, the "percentile" bootstrap behaves correctly across almost all $2 \times 228$ experiments and across all test set sizes. Note that all outliers with very high coverage actually correspond to distributions containing a Dirac weighing more than half the total distribution mass. In such cases, overcoverage is a normal behavior.

## 4   Discussion

To our knowledge, this work presents the first systematic comparison of CI methods for medical image segmentation metrics. To that purpose, we performed an extensive analysis across 228 experiments covering a variety of tasks and segmentation methods.

Reporting CIs is crucial, both for proper validation of methods and to align with clinical and regulatory guidelines [7] which is key for translation. However, there are yet no guidelines on how to compute CIs in practice in medical imaging AI. In this preliminary study, we focused on coverage properties and two common performance metrics for image segmentation. We unveiled some major pitfalls. Critically, the choice of the CI method depends upon the summary statistic: mean vs median (or any other order-based statistic).

The most striking results are for CIs of the median, where "basic" and "BCa" bootstrap led to some catastrophic failures while "percentile" was adequate across all experiments. We believe that these results are important. Median is a common robust statistic for central tendency. Many users can be tempted to blindly rely on "BCa" or "basic" because they are often recommended over "percentile" and because "BCa" is the default in some bootstrap implementations. For CIs of the mean, the results are less surprising, but there are still some interesting lessons. All methods behave decently when $n$ is large enough but "BCa" and "parametric t" offer a substantial advantage when $n$ is small. Perhaps more surprising is that some experiments exhibit poor coverage across all methods, even for relatively large $n$. We inspected the metric distributions corresponding to these benchmarking instances and found that they either have major outliers or concentrate large mass in a single point. Thus, in such cases, the mean alone can be misleading and adequate CIs can require a large sample size.

The failures of "BCa" when used for CIs of the median can be explained by the way the so-called acceleration factor corrects for skewness using a jackknife estimate. Indeed, the denominator of this factor is the variance of leave-one-out estimates of the statistic which can easily become zero or very small for the median (or for any order statistic). The jackknife failure for non-continuous summary statistics is thus natural when one looks at the formula and has indeed been described [1], but we believe it is not common knowledge. The curious reader may have noticed that "BCa" is worse when $n$ is even than when $n$ is odd. Again, this can be explained from the acceleration formula: the leave-one-out estimates of the median are more likely to be equal when $n$ is even. The "basic" bootstrap has been debated, with some arguing that it provides some bias correction while others say that it is "asymmetric in the wrong direction for skewed data" [12]. Indeed, our results show that for segmentation metrics, where skewness is highly common, "basic" is the worst method both for mean and median.

Availability in software packages is likely a driving factor underlying researchers' choices, and the AI community is almost entirely Python-based. It is extremely useful that the function bootstrap is available in SciPy [19], because it "democratizes" the use of this statistical methodology. The function implements

the three most common versions which are "basic", "percentile" and "BCa", the default method being "BCa". Our results show that there is no "one-size-fits-all" method. We thus invite the medical imaging community to carefully consider the available options when using the function: going beyond the default is easily overlooked.

There is very little work on CIs for validation metrics in medical imaging. Typical values of CI width have been reported [10] but without assessing coverage which is fundamental to know whether a CI method is adequate. Other studies provide insights about image-derived quantities [4,21,17], but not for validation metrics. This lack of literature in medical imaging is in sharp contrast with other fields including psychology [13], neuroscience [18], social sciences [14] and generalist AI [2,15]. In general, these works do not come to the same conclusions as we do which can be explained by major differences: some other works rely on parametric simulations which are inadequate for segmentation metrics, some look only at the mean, some deal with cases where $n$ is very large. We thus believe that medical imaging requires specific studies on the behavior of CIs.

This work has the following limitations. So far, we have only studied the case of segmentation and two metrics, and future work will need to tackle other tasks and other relevant metrics. In particular, the case of metrics with discrete or unbounded support is important and is left for future work. Moreover, due to space constraints, we focused on CI coverage. CI width is another important property that guides selection of a CI method among those which provide adequate coverage.

Even though preliminary, our results have immediate consequences on how to compute CIs for medical image segmentation. Indeed, we unveiled cases where some CI methods behave catastrophically which should have immediate impact on researchers reporting practices. Our work contributes towards the creation of standardized guidelines on CI reporting in medical imaging AI.

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
