# OpenReview forum: "Some hidden traps of confidence intervals in medical image segmentation: coverage issues"
_MICCAI.org/2025/Workshop/BRIDGE — BRIDGE 2025 Poster_

### Official Review · Reviewer_qusT · 2025-07-23
**Empirical Evaluation of Confidence Intervals in Medical Image Segmentation**

**Rating:** 7
**Confidence:** 5

**Review:**

This paper presents a methodologically rigorous study of confidence interval (CI) methods for evaluating segmentation performance in medical imaging, with a focus on empirical coverage properties. The authors conduct an extensive non-parametric simulation study across benchmarking instances derived from the Medical Segmentation Decathlon, using realistic sample size regimes and metrics that reflect common practice. Their methodological approach is sound, particularly the use of kernel density estimation.

The writing is generally clear, and the exposition is logically structured. Key statistical concepts, such as theoretical versus empirical coverage, are defined with precision, and the rationale for the experimental design is well articulated. The figures support the analysis effectively. The paper makes an original contribution by systematically evaluating CI methods in a domain where such scrutiny has been absent. The discovery that commonly used bootstrap methods, such as BCa and basic, can exhibit catastrophic undercoverage for median estimation is especially impactful.

The study has practical significance, aligning with evolving regulatory expectations that emphasize uncertainty quantification in AI validation. However, the paper's scope is somewhat limited by its restriction to segmentation and small number of metrics metrics, and it does not address other important properties of CIs such as interval width, which also influence the choice of method in practice. Furthermore, although the explanation of the BCa method's failure is plausible and rooted in the instability of the jackknife variance for order statistics, a more formal treatment of this issue would strengthen the argument.

Overall, this work is a valuable and well-executed contribution that addresses a gap in the literature. It is likely to influence both future research directions and practical reporting standards in medical imaging AI.

---

### Official Review · Reviewer_UcWY · 2025-07-24
**Review Comments**

**Rating:** 7
**Confidence:** 4

**Review:**

You have systematically compared five CI methods on 228 segmentation benchmarks from the Medical Segmentation Decathlon, and demonstrate when common techniques break down, especially for median estimates where basic and BCa bootstraps can fail badly, while the percentile bootstrap remains solid. That said, a few areas need more work.
1.	The choice and tuning of your kernel density estimator deserve justification. You use an Epanechnikov parabolic kernel with adaptive bandwidth to model metric distributions, but it’s unclear how you picked bandwidth parameters or assessed KDE quality. I would like to see a sensitivity analysis showing how different kernels or bandwidths affect coverage results to ensure that your findings aren’t an artifact of one setup.
2.	Reproducibility details are sparse. Please specify the random seeds, the exact SciPy versions you ran (“bootstrap” defaults can change), and the code for your KS goodness-of-fit tests. Sharing your scripts or a GitHub link would help reproducibility. Right now, I worry that slight differences in implementation could flip a method from “safe” to “catastrophic.”
3.	While Dice and NSD are widely used, segmentation validation also relies on other metrics like Hausdorff distance, volumetric errors, boundary F1 scores, especially in clinical settings. It would strengthen your message to include at least one additional metric or explain why Dice/NSD covers the most important cases. That way your recommendations apply more broadly, not just to two numbers.
4.	You could note that your simulations assume independence across samples, which may not hold if test cases share pathology subtypes, and that CI width is also critical for interpretation but was left for future study. Pointing to these next steps will guide the community toward complete reporting standards.

---

### Official Review · Reviewer_Ux3p · 2025-07-25
**Reviewer's Comments**

**Rating:** 9
**Confidence:** 4

**Review:**

### 1. Summary of the Paper
This paper examines whether different methods for calculating confidence intervals actually work well for medical image segmentation metrics like the Dice score and surface distance measures. The authors run extensive computer simulations across 228 different test cases to see how often these confidence interval methods give accurate results for both average and median values.

### 2. Strengths
* A very comprehensive evaluation of CI coverage for segmentation metrics across a large and diverse set of benchmarks.
* The authors address a very important methodological gap that affects clinical translation and regulatory alignment.

### 3. Limitations or Areas for Improvement
* The paper would benefit from reporting average confidence interval widths for each method across different test set sizes, along with an analysis of how coverage accuracy relates to interval width.

* While the discussion section identifies limitations in SciPy's implementation, it would be valuable to provide practical alternatives or recommendations for practitioners. The authors might consider making their code publicly available to help bridge this methodological gap for the research community.

### 4. Relevance to BRIDGE Workshop Topics
The paper looks at whether the statistical tools that researchers rely on to measure model performance are actually reliable. The authors show how confidence intervals, which are often used incorrectly or ignored completely, can impact whether medical AI systems get approved for clinical use and meet regulatory standards.

---

### Decision · Program_Chairs · 2025-07-25

**Decision:**

Accept (Poster)

**Comment:**

Dear Authors,

Congratulations!

We are pleased to inform you that your paper has been accepted for the BRIDGE Workshop. This paper addresses a critical gap in the reporting and evaluation of medical AI technologies.  Your paper was reviewed by three scientists from regulatory, academic, and industry backgrounds to provide different perspectives, and all agree it offers an important analysis and framework needed to translate medical AI from research into clinical practice.

Requirements for your final camera-ready submission (due July 30):
* Incorporate reviewer comments and suggestions where appropriate throughout your paper. At minimum, add a discussion section that acknowledges and responds to the key points raised by reviewers
* Ensure your final draft follows standard MICCAI conference formats and guidelines
* Please submit your camera-ready source file, and any supplementary material you might have.

We look forward to your presentation and the discussions it will generate at the workshop!

Best regards,
BRIDGE Workshop Organizers